# Health-related quality of life implications of plantar ulcers resulting from neuropathic damage caused by leprosy: An analysis from the trial of autologous blood products (TABLE trial) in Nepal

**Naomi Kate Gibbs**[1]*, **Jessica Ochalek**[1], **Indra Bahadur Napit**[2], **Dilip Shrestha**[2], **Pedro Saramago Goncalves**[1], **Richard J. Lilford**[3], **Mark Sculpher**[1]

1 Centre for Health Economics, University of York, York, United Kingdom, 2 Anandaban Hospital, The Leprosy Mission Nepal, Kathmandu, Nepal, 3 Institute of Applied Health Research, University of Birmingham, Birmingham, United Kingdom

* naomi.gibbs@york.ac.uk

**Data Availability Statement:** The data underlying the findings would require participant level trial

## Abstract

### Introduction

Leprosy is a curable disease, treated by multidrug therapy. However, patients are often left with neuropathic damage leading to lifelong vulnerability to ulcers. Quantifying the value of interventions to improve ulcer healing is challenging as data on the health impact of plantar ulcers are scarce, especially in low- and middle-income countries. We aim to quantify the impact of plantar ulcers on patients' health-related quality of life (HRQoL) using methods which can inform decisions about the effectiveness and cost-effectiveness of alternative forms of management.

### Methods

Generic HRQoL data were collected using the EuroQol (EQ)-5D-3L questionnaire in a randomised control trial in Nepal, treating plantar ulcers with leukocyte and platelet-rich fibrin or usual care. The trial followed 130 patients resulting in 600 observations. EQ-5D data were converted into a single 'utility' score by weighting the instrument's different dimensions using the preferences of the Sri Lankan population. Utility data were analysed using general estimating equation regressions. The impact of an ulcer on HRQoL was estimated whilst controlling for clinical and demographic covariables.

### Results

Estimated mean utility (standard error) for the sample across all time points was 0.52 (0.02) for patients with an ulcer and 0.64 (0.01) with a healed ulcer. Controlling for clinical and demographic covariates, we estimate that the presence of an ulcer leads to a 0.12 (0.02) decrement in HRQoL compared with a healed ulcer.

data being made publicly available which could compromise patient privacy, and participants did not consent for their data to be made publicly available. We are however able to share non-identifiable participant level data upon request with an appropriate data sharing agreement in place at following email: bctudatashare@contacts.bham.ac.uk. For more information, please click at the link: https://www.birmingham.ac.uk/research/bctu/data-sharing-and-protection-policy.

**Funding:** This research was funded by the National Institute for Health Research (NIHR) (NIHR200132) using UK aid from the UK Government to support global health research. RJL is also funded by NIHR Applied Research Collaboration (ARC) West Midlands and Midlands Patient Safety Research Collaboration (PSRC). The views expressed in this publication are those of the author(s) and not necessarily those of the NIHR or the UK Department of Health and Social Care.

**Competing interests:** The authors have declared that no competing interests exist.

## Discussion

Based on the levels of health they report and the values of the general public, patients with a history of leprosy are at risk of significant HRQoL burden from neuropathic plantar ulcers. Quantifying this health impact provides important evidence to assess the effectiveness and cost-effectiveness of leprosy ulcer interventions and ensures that interventions for leprosy ulcers can be appropriately considered for funding under Universal Health Coverage against other potential uses of the same money.

## Introduction

Leprosy is an infectious mycobacterial disease affecting the skin, nerves, respiratory tract and eyes. It is curable with multidrug therapy and, if caught early, patients may be free of ongoing health issues. However, a third of newly diagnosed patients have nerve damage with a consequent lifetime risk of recurring ulcers and other disabilities [1]. Leprosy is associated with poverty, low levels of education, and malnutrition, with patients often living in deprived rural communities and relying on subsistence roles [1–3]. Disability resulting from leprosy can be stigmatising and result in increased economic challenges for the patient [4].

Nepal is one of the World Health Organisation's (WHO) global priority countries for their Global Leprosy Strategy 2021–2030 due to prevalence and new case detection, estimated at 3249 in 2018 [5, 6]. Beliefs regarding leprosy resulting from former life transgressions and associated stigma in Nepal can prevent patients seeking treatment resulting in ulcers and the requirement of inpatient hospital care [7]. Plantar ulcers are those which occur on the underside of the foot. Plantar ulcer healing requires reducing the pressure on the ulcer but due to the physical work requirements of many of the patients this is impractical creating a cycle between poverty and ill health. Treatment of plantar ulcers caused by neuropathic damage includes debridement, controlling infections, reducing pressure areas, optimising blood flow and nerve decompression [8]. Medical trials seeking to explore optimal care for ulcer patients include self-care, footwear, education, laser therapy and various applications during dressing changes [9]. The efficacy of such treatments may be measured using disease-specific clinical endpoints, such as the rates of healing and of recurrences. However, to inform decisions regarding the funding of treatments under Universal Health Coverage (UHC), reliable health-related quality of life (HRQoL) data are needed which are comparable across disease areas (i.e., they are *generic*). These data can be used as part of cost-effectiveness analyses of interventions against other potential uses of the healthcare budget (i.e., for treatments in other disease areas).

A generic health questionnaire asks the patient about their overall level of health over a range of areas, such as pain and discomfort, anxiety and depression and their ability to engage in usual activities. The answer to these questions defines the health "state" that each patient is in. These health states can then be expressed as a single value t, known as a utility value, based on interviews with a sample of patients or the general public regarding their preferences between the health states [10]. The utility value is zero for death and one for perfect health, scores less than zero represent a health state worse than death. If a health state was valued at 0.60 then it indicates that, on average, the public would be willing to forgo approximately 40% of their future life expectancy to avoid this health state [10]. Studies which focus on LMICs, and the Institute for Health Metrics and Evaluation (iHME), will often use the concept of

disability weights instead. The principles are very similar but, in this case, zero represents perfect health and one represents dead.

There is little evidence on the HRQoL for patients with neuropathic damage caused by leprosy but who no longer have leprosy. The iHME provide disability weights for disfigurements due to leprosy using regression methods applied to general population surveys [11]. One systematic review has estimated the HRQoL impact of disfigurement due to leprosy applying a meta-analysis to SF-36 generic health questionnaires completed by leprosy patients receiving treatment [12].

To contribute to the evidence on the long-term health consequences of this neglected tropical disease, a recent individual randomised efficacy trial of autologous blood products to promote ulcer healing, in patients with neuropathic damage caused by leprosy, in Nepal collected EQ-5D data for all patients across multiple time points. Our objectives in this paper are to estimate the utilities for patients with neuropathic damage caused by leprosy and to estimate the specific health-related quality of life decrement associated with having a simple ulcer, whilst accounting for relevant covariables.

## Methods

### TABLE trial

This analysis uses EQ-5D data collected as part of the TABLE trial, full details of which have previously been reported [13]. In summary, TABLE recruited 130 eligible patients presenting to the Anandaban Hospital in Nepal with a plantar ulcer resulting from neuropathic damage, from September 2020 to May 2022. Patients were excluded from the trial if they had significant medical conditions such as diabetes, HIV, chronic HEP B, chronic HEP C or TB, full exclusion criteria are provided in the trial results [13, 14]. Patients did not enter the trial unless their ulcer was defined as a simple ulcer (one in which a probe cannot touch the bone). Patients received either standard care, in which they received twice weekly saline dressing changes as inpatients of the hospital, or the Leukocyte and Platelet Rich Fibrin (L-PRF) intervention, which additionally used the patient's centrifuged blood to obtain a natural fibrin matrix gel which is placed over the ulcer during the dressing changes. Patients remained in the trial until 70 days after randomisation or ulcer healing, whichever was first, and were followed up at six months post randomisation. The trial's primary outcome was rate of ulcer healing and no statistically significant effect of L-PRF versus standard care was identified.

At each dressing change, the ulcer was assessed for healing. There were two assessments, one by the local clinician and another by a blinded assessor using photos of the ulcer. For consistency with the clinical analysis, we use the blinded assessments for our analysis.

### Utility data

Patients in the trial completed the EQ-5D-3L. This involves them indicating their level of health (1 = no problems, 2 = some problems, 3 = extreme problems) on five domains within this measure (mobility, self-care, usual activities, pain/discomfort, anxiety/ depression). This questionnaire is designed to capture patients' overall level of health, not just the impact of a specific illness, in this case the ulcer. These five levels are then transformed into a single utility value using the preferences elicited from a sample of the general public in Sri Lanka, as there is currently no value set for Nepal and Sri Lanka was considered by the clinical experts to be the most similar country in which a preference study had been undertaken [15].

EQ-5D data were collected at the point of randomisation and at two weekly intervals thereafter, up to the point that the patient's ulcer healed (at which point they exit the trial)

or 70 days, whichever comes first. EQ-5D data were also collected six months after randomisation (Fig 1).

## Analysis

Baseline clinical and demographic characteristics of the sample are summarised as well as the number of EQ-5D observations at each time point. Summary data across the five EQ-5D domains are reported for patients with and without an ulcer. Mean utility scores are estimated and plotted, disaggregated by treatment group.

In order to estimate the health decrement resulting from a simple ulcer versus a healed ulcer, we transformed patient-level utility scores into utility decrements by subtracting them from one (perfect health), creating a non-negative distribution. We ran General Estimating Equations (GEE) regressions to account for both the distributional characteristics and the repeated measures characteristics of the data [16–18]. Different model specifications were explored considering several correlation structures, family, link and dependent variables. We included all demographic and clinical characteristics available in the data [13], removing hypertension as it overlapped with the blood pressure measures (Eq (1)). The only time-varying covariable was whether or not the ulcer is healed, all other variables were measured only at baseline. We assessed the models using the QIC measure of fit, QQ plots and the clinical/face validity of the results.

$$g(disutility_{ij}) = \beta_0 + \beta_1 Ulcer_{ij} + \beta_2 Female_{ij} + \beta_3 Age_{ij} \ldots \ \ldots \ \ldots \ \ldots \beta_{19} X_{ij} + \varepsilon_{ij} \qquad (1)$$

*g(.) is a specified link funtion (e.g. identity, log) which estimates disutility*

*for the $i^{th}$ individual at the $j^{th}$ time point*

*the family was also specified (i.e. gamma, gaussian)*

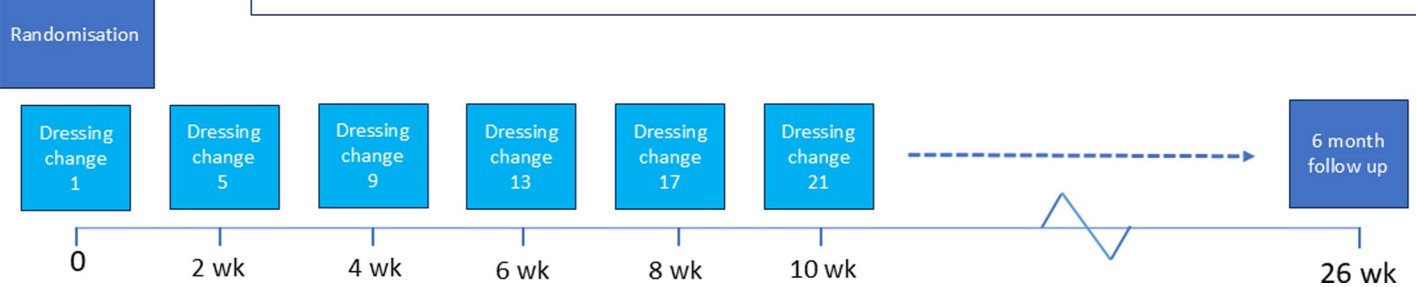

**Fig 1. Schematic describing the TABLE trial protocol for EQ-5D data collection.**

### Ethics

This study was approved in the UK by the University of Birmingham Biomedical and Scientific Research Ethics Committee (BSREC ERN 19–1960) and locally in Nepal through the Nepal Health and Research Council (NHRC ERB 303/2020 P). Participants were recruited between 15th September 2020 and 24th May 2022. A patient information leaflet was provided and the potential participant was invited to sign a consent form on the following day. Local clinically trained researchers obtained written consent from all participants. Patient information leaflet and consent forms were translated from English and back-translated according to the WHO methodology for high fidelity translation. The obtaining of informed consent will be evidenced by the completion of an electronic case report form (e-CRF) within the REDCap closed data capture system by staff named and on the delegation log.

## Results

### Description of the sample

Baseline clinical and demographic characteristics are presented (Table 1). The majority of patients are men, with only 20% female. The mean age of patients is 54 years. Of all patients

**Table 1. Clinical and demographic characteristics at baseline.**

|  | Overall (n = 130) | Means for continuous variables (sd) |
|---|---|---|
| Female | 27 (20%) |  |
| Age* |  |  |
| 18–39 | 29 (22%) |  |
| 40–59 | 49 (38%) |  |
| 60+ | 52 (40%) |  |
| Age (in years) |  | 54.03 (15.77) |
| Body Mass Index |  | 22.48 (3.53) |
| Systolic blood pressure (mmHg) |  | 113.46 (10.91) |
| Diastolic blood pressure (mmHg) |  | 77.62 (7.95) |
| Platelet result (x10^3/ul) |  | 283.71 (88.38) |
| Haemoglobin (gm/dL) |  | 13.65 (1.87) |
| Fasting blood sugar (mg/dL) |  | 84.38 (11.79) |
| Years since leprosy diagnosis |  | 19.73 (14.75) |
| Currently taking antibiotics |  |  |
| Ongoing | 11 (9%) |  |
| Completed | 118 (91%) |  |
| Unknown | 1 (0%) |  |
| Hypertension or hyperthyroidism | 20 (15%) |  |
| Nerve enlarged in either leg | 76 (56%) |  |
| Loss motor function in either foot | 66 (51%) |  |
| Deformity in foot | 117 (90%) |  |
| More than one plantar ulcer | 27 (21%) |  |
| Weeks the trial ulcer has been unhealed |  | 50.75 (88.67) |
| Trial ulcer is a recurrent ulcer | 81 (62%) |  |
| Baseline ulcer area cm$^2$ (measured using the PUSH tool) |  | 3.81 (2.88) |

* The age categories are derived from the continuous age variable. The continuous age variable is given a best estimate where it is not known, this is true for 56 patients in the sample

entering the study, 90% had a foot deformity at baseline and for 62% of patients the trail ulcer was a recurrent one.

All 130 patients who entered the trial with a simple ulcer report their EQ-5D at baseline. As patients heal, they leave the trial and so the number of observations as time passes reduces. There was no missing data until the six-month follow up, when observations for 5 patients (4%) were missing (Table 2). Across all patients and time points, there were 600 EQ-5D observations from the TABLE trial. Only four patients in the trial had an ulcer that did not heal by six months, although many patients had an ulcer that healed during the trial but had then recurred by six months (ref clinical paper).

## HRQoL analysis

Patients' responses across the three levels and five domains of the EQ-5D are reported separately for observations when patients have an ulcer versus a healed ulcer (Table 3). Across all domains, there are higher proportions reporting Level 1 (no problems) amongst patients with healed ulcers. The domain with the biggest discrepancy between those with an ulcer and with a healed ulcer is self-care, followed by mobility.

The unadjusted means for all observations with a simple ulcer and a healed ulcer are 0.52 (SE = 0.01) and 0.64 (SE = 0.02), respectively. Mean utility between treatment groups was higher for LPRF patients but this was not statistically significant (Fig 2).

## Modelling utility decrements

The best model for disutility, assessed using the QIC and visual inspection of the QQ plots, assumed the gaussian family and identity link. Alternative specifications, namely the gamma family using identity link and gaussian using log link, are provided (S1 Table and S1 Fig). The correlation structure chosen, using the QIC, was exchangeable on inspection of residuals between time points following the fitting of a simple linear model (S2 Table). The results provide estimates of the disutility whilst controlling for covariates (Table 4). The intercept can be interpreted as the disutility if all covariates are set to zero; however, as many of the covariates cannot be zero in this context (i.e. age, years since leprosy diagnosis) it is not perfectly interpretable. Despite this, since the coefficients are very small, except for ulcer and female, we can broadly interpret the constant as the disutility for males with no ulcer, in this patient group. This implies a HRQoL score of 0.65 (a score of 1 = full health) which demonstrates these patients are experiencing health problems even without accounting for the ulcer. Having an ulcer has the most significant and largest magnitude of impact on disutility with a 0.11 (SE = 0.02) decrement. Being a female has the next largest impact, 0.09 (SE = 0.04), and is significant. Baseline age is also significant with a small decrement for each additional year of age. Finally, BMI and having more than one ulcer at baseline are marginally significant. Seven

**Table 2. Number of observations at each time point.**

| Time point | Number of observations |
|---|---|
| Randomisation | N = 130 (100%) |
| 2 weeks | N = 127 (98%) |
| 4 weeks | N = 97 (75%) |
| 6 weeks | N = 63 (48%) |
| 8 weeks | N = 34 (26%) |
| 10 weeks | N = 24 (18%) |
| 6 month follow up | N = 125 (96%) |

**Table 3. The number of observations (% share) across the three levels for each EQ-5D domain.**

| Domain | Level* | Ulcer (obs = 477) | Ulcer (%) | Healed ulcer (obs = 123) | Healed Ulcer (%) |
|---|---|---|---|---|---|
| Mobility | 1 | 147 | 30.8% | 71 | 57.7% |
| | 2 | 329 | 69.0% | 52 | 42.3% |
| | 3 | 1 | 0.21% | 0 | 0% |
| Self-care | 1 | 234 | 49.1% | 99 | 80.5% |
| | 2 | 237 | 49.7% | 23 | 18.7% |
| | 3 | 6 | 1.3% | 1 | 0.81% |
| Usual activities | 1 | 304 | 63.7% | 98 | 79.7% |
| | 2 | 168 | 35.2% | 22 | 17.9% |
| | 3 | 5 | 1.1% | 3 | 2.4% |
| Pain/discomfort | 1 | 146 | 30.6% | 54 | 43.9% |
| | 2 | 242 | 50.7% | 50 | 40.7% |
| | 3 | 89 | 18.7% | 19 | 15.4% |
| Anxiety/Depression | 1 | 146 | 30.6% | 54 | 43.9% |
| | 2 | 242 | 50.7% | 50 | 40.7% |
| | 3 | 89 | 18.7% | 19 | 15.4% |

* 1 = no problems, 2 = some problems, 3 = severe problems

observations were dropped, all belonging to a single patient, as the ulcer area measure at baseline was missing.

## Discussion

Using all of the EQ-5D observations from the TABLE trial, we have provided the first estimates, to the best of our knowledge, for the utility of patients with neuropathic leprosy damage resulting in plantar ulcers, both with an ulcer (0.52, SE = 0.02) and with a healed ulcer (0.64,

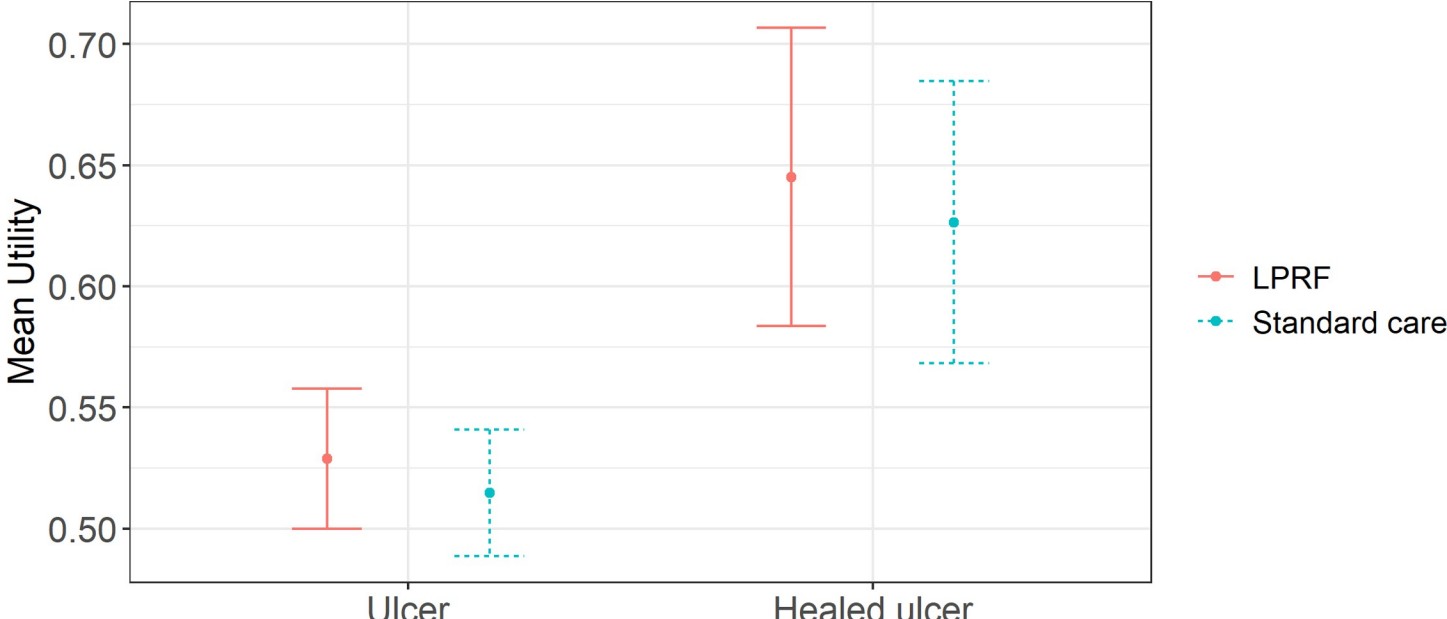

**Fig 2. Mean utility for TABLE trial patients with an ulcer versus a healed ulcer, split by treatment group, with 95% confidence intervals.**

**Table 4. GEE GLM results with disutility as dependent variable, obs = 593.**

|  | Estimate | Std. Error | Pr(>\|W\|) |
|---|---|---|---|
| Intercept | 0.35 | 0.23 | 0.13 |
| Ulcer | 0.11 | 0.02 | 0.00*** |
| *The following variables were measured at baseline only* |  |  |  |
| Female (male = reference) | 0.10 | 0.04 | 0.01* |
| Age | 0.00 | 0.00 | 0.02* |
| BMI | 0.01 | 0.00 | 0.09. |
| Systolic bp | 0.00 | 0.00 | 0.12 |
| Diastolic bp | 0.00 | 0.00 | 0.59 |
| Platelet result | 0.00 | 0.00 | 0.70 |
| Haemoglobin | 0.00 | 0.01 | 0.69 |
| Fasting blood sugar | 0.00 | 0.00 | 0.98 |
| Years since leprosy diagnosis | 0.00 | 0.00 | 0.95 |
| Antibiotics (ongoing = reference) |  |  |  |
| Completed | -0.05 | 0.06 | 0.40 |
| Unknown | -0.04 | 0.07 | 0.61 |
| Nerve enlarged in either leg | 0.00 | 0.03 | 0.89 |
| Loss motor function in either foot | 0.02 | 0.03 | 0.38 |
| Deformity in foot | -0.03 | 0.04 | 0.43 |
| More than one ulcer | 0.06 | 0.04 | 0.10. |
| Weeks the trial ulcer has been unhealed | 0.00 | 0.00 | 0.87 |
| Trial ulcer is a recurrent ulcer | 0.04 | 0.03 | 0.18 |
| Ulcer area cm2 (measured using the PUSH tool) | 0.00 | 0.00 | 0.41 |

Significance codes: 0 '***' 0.001 '**' 0.01 '*' 0.05 '.' 0.1 ' ' 1

Family = gaussian

Link = identity

Correlation structure = exchangeable

Observations = 593

Number of clusters: 129 Maximum cluster size: 7

SE = 0.01). Estimating disutility using a GEE model, the health decrement associated with an ulcer is 0.11 (SE = 0.02).

We compare our results with two other studies; the iHME global burden of disease [11] and a recent systematic literature review and meta-analysis of leprosy disability weights [12]. The iHME produce disability adjusted life year weights derived from surveys of the general population across multiple countries (including Hungary, Italy, the Netherlands, Sweden, Bangladesh, Indonesia, Peru, Tanzania, USA), using pairwise comparisons and regression models to generate weights between zero and one, where zero equals full health and one is equivalent to death [11, 19]. They provide two values for disfigurement due to leprosy, dependent on the level of disfigurement. Level one is defined as having a slight, visible physical deformity that others notice, which causes some worry and discomfort and is estimated at 0.01 (CI = 0.01–0.02). Level two is defined as having a visible physical deformity that causes others to stare and comment. As a result, the person is worried and has trouble sleeping and concentrating and is estimated at 0.07 (CI = 0.04–0.10).

There is no specific reference to ulcers in the iHME weights relating to leprosy. Therefore, another useful point of comparison are ulcers resulting from other conditions. The iHME have generated disability weights for decubitus ulcers (pressure ulcers or bedsores) which have

a very similar description in terms of visibility and resulting worry, as listed in level one and two for disfigurement due to leprosy, but with the added characteristics of itching and soreness (0.027/0.188/0.576 mild/moderate/severe). The severe state includes avoiding social contact and considering suicide. Our estimate of the decrement associated with a simple ulcer (0.11) is higher than both of the iHME disability weights related to disfigurement due to leprosy, and lies between the mild and moderate estimates for pressure ulcers.

A systematic review by Nanjan Chandran et al. (2021) suggests the iHME disability weights for disfigurement due to leprosy are too small, underestimating the health burden. In their meta-analysis, they use individual patient data sourced from 667 individuals across eight studies, which reported SF-36, to estimate leprosy disability weights by grade of disability. Grade 0 is defined as an absence of disability and no visible damage or deformities on eyes, hands and feet; grade 1 as loss of protective sensibility in the eyes, hands or feet, but no visible damage or deformities; grade 2 as presence of deformities or visible damage to the eyes, visible damage on hands or feet (hand with ulcerations and/or traumatic, resorption, claw, fallen hand, ulcers; feet with trophic and/or traumatic injuries, resorption, claw, foot drop, ulcers, ankle contracture). The overall estimated disability weight for grade 2 disability was 0.26 (95% CI: 0.18–0.34). For grade 1 was 0.19 (95%CI: 0.13–0.26) and for grade 0 was 0.13 (95%CI: 0.06–0.19). A simple ulcer as defined in our study would fall under the grade 2 definition, our estimated decrement, at 0.12, is smaller than that of Nanjan Chandran, Tiwari [12]. This difference could be driven by several methodological and empiric factors. Firstly, their review includes studies which collected SF-36 data whereas we use the EQ-5D, different generic based measures of health. We also have longitudinal data providing observations from the same patients when they have the ulcer and when they do not. This allows us to estimate the health-related quality of life impact of the ulcer more accurately. Their meta-analysis focused on patients with an active leprosy infection, whereas our study focuses on patients who have been cured of leprosy, but who are living with the long-term effects of the neuropathic damage resulting from the infection. Our study only includes patients with simple ulcers, however the grade 2 disability definition used by Nanjan Chandran, Tiwari [12] is much broader and includes health complications which may have a greater HRQoL impact.

Although there are methodological differences between the studies, our analysis accords with Nanjan Chandran et al. (2021) in suggesting that the iHME weights for leprosy disfigurement may underestimate the long-term health impact caused by leprosy. This is potentially due to the oversimplified description of the health states and the failure to account for how a condition impacts on dimensions such as usual activities [20].

Our finding that the utility for women within our sample is 0.09 (SE = 0.04) lower than males, using the disutility model, corresponds with previous research that found women who develop leprosy are especially disadvantaged, with rates of late diagnosis and disability high in this subgroup (Rodrigues and Lockwood, 2011). However, our results contradict Joseph and Rao [21] who reported higher HRQoL in females than males amongst a sample of patients released from leprosy treatment in India. While this study reports overall scores that are very similar between males and females, they do not report standard errors so we are unable to evaluate whether this difference was statistically significant. They also only have 50 data points. We believe our study provides an updated and more robust estimate given both the data and the methods employed.

Our study has some limitations. Our EQ-5D data are drawn from a trial based at an NGO-run hospital in Nepal, and includes only 130 patients, which may limit generalisability, although the study did provide longitudinal data with 600 observations in total. We have also used a value set from Sri Lanka to transform the levels reported within each EQ-5D domain to a single utility value per observation, as there is no value set available for Nepal. While there

may be important differences between Sri Lanka and Nepal, we believe this set is the best choice of the validated tariffs available [22]. The value set in Sri Lanka has been compared to those from high-income settings and a high level of agreement found on utilities reported for better health states. However, as health states get worse there are important differences, particularly relating to mobility. The Sri Lankan public rated problems with mobility as more important than high-income country value sets which might reflect the impact of disability on functioning in low-income settings (Kularatna et al. 2015). The indirect estimation of health utility values, as used by our study, can result in lower health ratings than if the utility had been estimated directly from the population [23]. Self-reported health-related quality of life measures are subject to many forms of bias, which may also vary depending on the cultural context. One form of bias is collection mode related bias in which patients may provide responses according to what they believe researchers would like to collect, it is possible this may be more prevalent in countries with a certain cultural norms [24, 25].

A further limitation is that our utility estimates are limited to simple ulcers as no ulcers became complicated in the trial. When ulcers develop to become complicated (defined as where a probe touches the bone) this can cause deep infection and non-healing ulcers which may lead to amputations. Utilities for complicated ulcers are even harder to estimate as the numbers of relevant patients are smaller. A pragmatic, though not ideal, approach could be to combine data sources, for example using a study where patients were asked to value various states related only to the presence of ulcers and amputations (Redekop et al. 2004). These additional decrements could then be anchored to the baseline utility values for patients with neuropathic leprosy damage estimated in our study.

Our study has provided estimates of the HRQoL for patients with neuropathic damage caused by leprosy both with an ulcer and with a healed ulcer. Previous research has provided estimates that we suggest may underestimate the health impact [11] or focus on patients with leprosy making the specific impact of the ulcer harder to identify. Our estimates, derived from multiple EQ-5D observations from the same individual, allow us to far better capture the health impact for this neglected patient group. This work is highly useful for informing economic evaluations, which requires that any health gains from leprosy ulcer interventions can be compared against other uses of the same money.

## Conclusion

Our study provides the first known estimates of health state utilities for patients with neuropathic damage caused by leprosy (but who have been cured of the leprosy infection) with a healed ulcer and those with a simple ulcer in a low-income setting. These can be used to analyse and inform future trials of interventions to improve healing and reduce recurrence of leprosy ulcers.

## Supporting information

**S1 Table. Comparing disutility model under different family and link assumptions.**
(DOCX)

**S2 Table. Correlation between residuals in linear model.**
(DOCX)

**S1 Fig. Comparing disutility models using QQ plots under different family and link assumptions.**
(DOCX)

## Author Contributions

**Conceptualization:** Naomi Kate Gibbs, Jessica Ochalek, Indra Bahadur Napit, Dilip Shrestha, Pedro Saramago Goncalves, Richard J. Lilford, Mark Sculpher.

**Formal analysis:** Naomi Kate Gibbs, Jessica Ochalek.

**Methodology:** Pedro Saramago Goncalves.

**Project administration:** Indra Bahadur Napit, Dilip Shrestha.

**Supervision:** Pedro Saramago Goncalves, Mark Sculpher.

**Visualization:** Naomi Kate Gibbs.

**Writing – original draft:** Naomi Kate Gibbs.

**Writing – review & editing:** Naomi Kate Gibbs, Jessica Ochalek, Indra Bahadur Napit, Dilip Shrestha, Pedro Saramago Goncalves, Richard J. Lilford, Mark Sculpher.

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
