## [Decision Letter · Decision Letter 0]

1 Nov 2024

PONE-D-24-19487Health-related quality of life implications of plantar ulcers resulting from neuropathic damage caused by leprosy: An analysis from the trial of autologous blood products (TABLE trial) in NepalPLOS ONE

Dear Dr. Gibbs,

Thank you for submitting your manuscript to PLOS ONE. After careful consideration, we feel that it has merit but does not fully meet PLOS ONE’s publication criteria as it currently stands. Therefore, we invite you to submit a revised version of the manuscript that addresses the points raised during the review process.

We look forward to receiving your revised manuscript.

Kind regards,

Claudia Sommer

Academic Editor

PLOS ONE

**Journal Requirements:**

This research was funded by the National Institute for Health Research (NIHR) (NIHR200132) using UK aid from the UK Government to support global health research. RJL is also funded by NIHR Applied Research Collaboration (ARC) West Midlands and Midlands Patient Safety Research Collaboration (PSRC). The views expressed in this publication are those of the author(s) and not necessarily those of the NIHR or the UK Department of Health and Social Care.

Reviewers' comments:

Reviewer's Responses to Questions

**Comments to the Author**

1. Is the manuscript technically sound, and do the data support the conclusions?

Reviewer #1: Partly

2. Has the statistical analysis been performed appropriately and rigorously? 

Reviewer #1: Yes

3. Have the authors made all data underlying the findings in their manuscript fully available?

Reviewer #1: Yes

4. Is the manuscript presented in an intelligible fashion and written in standard English?

Reviewer #1: Yes

5. Review Comments to the Author

**Reviewer #1:** This manuscript used multi-dimensional health outcomes which were reduced to a single index using health utilities based on a health-related quality of life questionnaire (EQ-5D) to enable broader comparisons of cost-utility or cost-effectiveness analysis of future interventions or treatments in patients with neuropathic damage caused by leprosy, specifically plantar ulcers. Conducting such studies is critical, as they provide empirical data that inform resource allocation, cost-effectiveness analysis, and policy-making decisions. By understanding public preferences and the trade-offs people are willing to make between quality of life and life expectancy, policymakers can develop strategies that more effectively align healthcare interventions with societal values and improve overall health outcomes.

In Introduction:

line 54 - It would be more appropriate to use 'mycobacterial' instead of 'bacterial' in this context, as it provides greater specificity and accurately reflects the genus of the pathogen involved in leprosy.

line 62 - Further clarification regarding the rationale behind selecting plantar ulcers as the primary burden, given that neuropathic damage in leprosy can present with numerous clinical manifestations (For example chronic neuropathic pain in leprosy treated patients) would provide more background to the reader. Additionally, it would be valuable to provide more regional epidemiological data on the prevalence and impact of plantar ulcers in Nepal to better contextualize this choice. This would enhance the understanding of the specific burden of plantar ulcers in comparison to other neuropathic complications associated with leprosy.

In Methods:

line 103 - Although it is noted that details are provided in a previous publication, given the significance of the topic, it would be important to clarify whether comorbidities that can also cause neuropathic damage, such as diabetes, physical injury (trauma), infections, autoimmune diseases, toxins, chemotherapy, medications, and other causes were excluded.

line 119 - During the EQ-5D questionnaire application, respondents were instructed to base their responses specifically on the impact of the plantar ulcer alone? Do the reported health-related quality of life outcomes accurately reflect the burden of the ulcer itself? Or other concurrent health conditions could influence their responses. (Please add any statement if needed in the limitations sections of the manuscript)

In Results:

Table 1 - The "more than one ulcer" variable would reflect ulcers in the same location (plantar or different parts of the body?)

Table 4 - It would be beneficial to include a clear explanation of what the intercept (first line of the table) represents in the math modeling, articulated in simple terms. This would enhance the accessibility of the results and ensure that readers without a strong statistical background can fully understand its significance in the context of the analysis.

In Discussion:

Is there any consideration of potential cultural factors that might influence respondents to report improvements in their condition as a form of respect or gratitude for receiving assistance? It would be useful to reference any published studies that have explored these social dynamics, particularly in similar healthcare contexts, as this could impact the accuracy of self-reported outcomes.

6. PLOS authors have the option to publish the peer review history of their article (what does this mean?). If published, this will include your full peer review and any attached files.

Reviewer #1: No

---

## [Author Response · Author response to Decision Letter 0]

2 Dec 2024

I have attached the response to reviewers word document

---

## [Editor Report · Decision Letter 1]

3 Dec 2024

Health-related quality of life implications of plantar ulcers resulting from neuropathic damage caused by leprosy: An analysis from the trial of autologous blood products (TABLE trial) in Nepal

PONE-D-24-19487R1

Dear Dr. Gibbs,

We’re pleased to inform you that your manuscript has been judged scientifically suitable for publication and will be formally accepted for publication once it meets all outstanding technical requirements.

Kind regards,

Claudia Sommer

Academic Editor

PLOS ONE
---

## [Editor Report · Acceptance letter]

30 Jan 2025

PONE-D-24-19487R1 

PLOS ONE

Dear Dr. Gibbs, 

I'm pleased to inform you that your manuscript has been deemed suitable for publication in PLOS ONE. Congratulations! Your manuscript is now being handed over to our production team.

Kind regards, 

on behalf of

Prof. Dr. Claudia Sommer 

Academic Editor

PLOS ONE